# Male Wistar Rats Chronically Fed with a High-Fat Diet Develop Inflammatory and Ionic Transport Angiotensin-(3–4)-Sensitive Myocardial Lesions but Preserve Echocardiographic Parameters

**DOI:** 10.3390/ijms252212474

**Published:** 2024-11-20

**Authors:** Thuany Crisóstomo, Rafael Luzes, Matheus Leonardo Lima Gonçalves, Marco Antônio Estrela Pardal, Humberto Muzi-Filho, Glória Costa-Sarmento, Debora B. Mello, Adalberto Vieyra

**Affiliations:** 1Leopoldo de Meis Institute of Medical Biochemistry, Center for Health Sciences, Federal University of Rio de Janeiro, Rio de Janeiro 21941-902, Brazil; thuany.crisostomo@bioqmed.ufrj.br; 2Graduate Program in Translational Biomedicine (BIOTRANS), Grande Rio University (UNIGRANRIO), Duque de Caxias 25071-202, Brazil; rafael.luzes@abeugraduacao.com.br; 3Grande Rio University (UNIGRANRIO), Duque de Caxias 25071-202, Brazil; matheusunigranrio@outlook.com; 4Carlos Chagas Filho Institute of Biophysics, Center for Health Sciences, Federal University of Rio de Janeiro, Rio de Janeiro 21941-902, Brazil; marco.pardal20@gmail.com (M.A.E.P.); humbertomuzi@biof.ufrj.br (H.M.-F.); sarmento@biof.ufrj.br (G.C.-S.); 5National Center for Structural Biology and Bioimaging/CENABIO, Center for Health Sciences, Federal University of Rio de Janeiro, Rio de Janeiro 21941-902, Brazil; debmello@cenabio.ufrj.br

**Keywords:** overweight/obesity, hypertension, high-fat diet, pro-inflammatory cytokines, cardiometabolic diseases, renin-angiotensin-aldosterone system, Angiotensin-(3–4), cardiac ion-transporting ATPases, echocardiography

## Abstract

The central aim of this study was to investigate whether male Wistar rats chronically fed a high-fat diet (HFD) over 106 days present high levels of interleukin-6 (IL-6) and tumor necrosis factor-alpha (TNF-α), and Na^+^ and Ca^2+^ transport alterations in the left ventricle, together with dyslipidemia and decreased glucose tolerance, and to investigate the influence of Ang-(3–4). The rats became moderately overweight with an expansion of visceral adiposity. Na^+^-transporting ATPases, sarco-endoplasmic reticulum Ca^2+^-ATPase (SERCA2a), and the abundance of Angiotensin II receptors were studied together with lipid and glycemic profiles from plasma and left-ventricle echocardiographic parameters fractional shortening (FS) and ejection fraction (EF). IL-6 and TNF-α increased (62% and 53%, respectively), but returned to normal levels with Angiotensin-(3–4) administration after 106 days. Significant lipidogram alterations accompanied a decrease in glucose tolerance. Angiotensin II receptors abundance did not change. (Na^+^ + K^+^)ATPase and ouabain-resistant Na^+^-ATPase were downregulated and upregulated, respectively, but returned to normal values upon Angiotensin-(3–4) administration. SERCA2a lost its ability to respond to excess ATP. Echocardiography showed no changes in FS or EF. We conclude that being overweight causes an increase in Ang-(3–4)-sensitive IL-6 and TNF-α levels, and ion transport alterations in the left ventricle that could evolve into future heart dysfunction.

## 1. Introduction

Among the processes and mechanisms that lead to severe comorbidities developing in obesity, including hypertension, coronary disease, and metabolic syndrome (dyslipidemia, type 2 diabetes), the renin angiotensin aldosterone system (RAAS) plays a significant role. It contributes to insulin resistance and stimulates inflammatory processes [1]. Research from our laboratory [2] has shown that overweight rats have a hyperactivated renal tissue RAAS with upregulation of Na^+^-transporting ATPases and significant interstitial fibrosis as a result of an early inflammatory process and expansion of visceral fat.

The central hypothesis of the present study was that cardiometabolic changes in young and moderately overweight hypertensive male rats receiving a high-fat, high-calory diet, can develop inflammatory cellular and molecular changes and some alterations in the population of colocalized Na^+^ and Ca^2+^ transporters in the left ventricle, potentially associated with hyperactivity of the local cardiac RAAS arm coupled to type 1 Angiotensin II (Ang II) receptors (AT_1_R). Based on the hypothesis that cardiometabolic changes in overweight rats involve AT_1_R, we aimed to investigate whether these changes are reflected in the echocardiographic structure and function of the heart, as well as the result of allosteric activation of the RAAS pathway coupled to type 2 Ang II receptors (AT_2_R) by Angiotensin-3–4 (Ang-(3–4)), a matter of interest since Ang-(3–4) is the antagonist of the AT_1_R-mediated Ang II effects in different physiological and pathological conditions [3,4].

## 2. Results

### 2.1. Development of Overweight and Hypertension in Wistar Hannover Rats on a High-Fat Diet

Figure 1 shows that rats that were fed the high fat diet (HFD) weighed an average of 259 g at age 58 days and had become overweight 106 days later compared to control (CTR) rats. The CTR diet was prepared following the recommendations of AIN-93 [5] containing 385 kcal/100 g of dry food with 9% of calories coming from lipids, and the rats reached an average body mass of 451 ± 53 g (mean ± SD). Meanwhile, the group exposed to the hypercaloric, high-fat diet (HFD) (574 kcal/100 g of dry food with 70% of calories coming from lipids) reached a body mass of 490 ± 61 g (mean ± SD), i.e., 10% greater.

Figure 2 compares the evolution of systolic blood pressure in CTR and HFD rats from exposure to the different diets until day 104. The recordings show that HFD rats became hypertensive from day 42 onwards. The administration of four doses of Ang-(3–4) (80 mg/kg BM) between days 104 and 106 normalized the systolic blood pressure in HFD rats (orange circle) but did not influence blood pressure in the CTR group (green circle) [6].

Figure 3 demonstrates that body mass remains unmodified when CTR and HFD rats that have not received Ang-(3–4) are housed in metabolic cages between day 104 and 106 (**A** and **B**). However, there is a significant decrease in the CTR group treated with Ang-(3–4) (**C**) but not in the HFD group (**D**). Water intake, urinary volume, and water balance were also analyzed 24 h after Ang-(3–4) administration.

Figure 4 shows that the administration of Ang-(3–4) causes an increase in water intake in CTR rats (A), which is accompanied by a proportionally similar increase in urinary volume over the same period (B). These changes in parallel result in an unmodified water balance in CTR rats receiving Ang-(3–4) (C). In the case of the HFD group, water intake is lower than in CTR (A), urinary volume (B) also has lower values that resulted in an unmodified water balance after administration of Ang-(3–4) (C). The comparison of studies on food and caloric intake [2], and water balance (the present work), allows us to conclude that housing rodents in metabolic cages alone or accompanied by the brief administration of Ang-(3–4) has an intrinsic variability whose causes remain open, as recently demonstrated [7].

### 2.2. Biomarkers of Metabolic Alterations and Inflammatory Processes in HFD Rats

In Figure 5, it can be seen that both epididymal fat (A) and perirenal fat (B) increased, respectively, 34 and 43% in HFD rats. Previous histological studies [2] show cell cross-sectional areas in adipocytes from epididymal and perirenal fat from CTR and HFD rats. Quantification of the areas demonstrates greater mean cross-sectional area per adipocyte, and area distribution analyses show an increase in larger adipocytes. The administration of four doses of Ang-(3–4) does not modify visceral fat levels around the two organs within 48 h only in HFD rats.

Figure 6 shows that the IL-6 levels in left-ventricular microsomes become 62% higher in HFD rats when compared to CTR animals (A and C). These figures show the opposite effects of Ang-(3–4): a return of IL-6 to CTR values in HFD rats and an increase of IL-6 found in HFD rats not treated with Ang-(3–4). When analyzing the TNF-α profile in the same microsomes, an increase of 53% in HFD rats is also found, with a return to CTR values in the HFD + Ang-(3–4) group and no influence in the CTR + Ang-(3–4) group (B and D).

### 2.3. Atypical Metabolic Syndrome in HFD Rats

Already at the time of the onset of hypertension, i.e., 70 days of exposure to the high-fat diet [2], HFD rats present a metabolic syndrome with a peculiar characteristic. Figure 7 shows that, although total cholesterol is similar in CTR and HFD groups (A), HFD rats have a marked decrease of 50% in high-density lipoproteins (B) and a mirrored increase in low-density lipoproteins (C). Total triglycerides are also reduced by just over 50% (D).

Disturbances in glucose metabolism also characterize metabolic syndrome in HFD rats. We measured blood glucose concentrations over time, and the curves allow us to infer whether there is glucose intolerance in HFD rats. They present a moderate but significant increase in fasting blood glucose (Figure 8A) and glucose intolerance (Figure 8B,C).

### 2.4. Unmodified Angiotensin II Receptors in Left-Ventricle Microsomes from HFD Rats

Figure 9A,C shows that the abundance of AT_1_R in left-ventricular microsomes is not modified in HFD rats compared to CTR rats. Moreover, Ang-(3–4) had no effect either. Figure 9B,D shows that there were no changes in the abundance of AT_2_R.

### 2.5. HFD Rats Present Alterations in the Active Transport of Na^+^ and Ca^2+^ in Left Ventricle That Are Reversed by Ang-(3–4)

The results below show the modifications of left-ventricular ATPases, i.e., primary active transporters of Na^+^ in the plasma membrane and transporter of Ca^2+^ across the sarco-endoplasmic reticulum membrane in HFD rats. Figure 10A shows that in rats that received the high-fat diet, there is a marked decrease in (Na^+^ + K^+^)ATPase activity, which was recovered in animals treated with Ang-(3–4). Administration of the peptide did not influence the pump from CTR rats. Figure 10B shows that the activity of the ouabain-resistant Na^+^-ATPase increases in HFD rats and normalizes after Ang-(3–4) administration. Figure 10C,D depicts data regarding (Na^+^ + K^+^)ATPase abundance in the membranes, demonstrating that it remains unmodified in HFD rats compared to CTR animals, without any influence of Ang-(3–4).

We investigated which type of commonly used equations, which describe the dependence of the rate of an enzymatic reaction on the substrate concentration, better describe the relationship between SERCA2a activity and ATP concentration [ATP] (Figure 11). For this, two simulations were carried out using the values of the average velocity for each [ATP] using two functions: a Michaelis-Menten function (Equation (1)) and another that considers the existence of inhibition due to excess substrate (Equation (2)), one of the most common deviations from Michaelian kinetics of physiological significance [10]. The velocity equations used were as follows.
V = (V_max_ × [ATP])/(K_m ATP_ + [ATP])(1)
V = (V_max_ × [ATP])/{(K_m ATP_ + [ATP]) × (1 + [ATP]/K_i ATP_)}(2)

Figure 11A shows the curves that best fit the respective means: in blue CTR, in pink HFD, in green CTR + Ang-(3–4), and in orange HFD + Ang-(3–4). The Michaelian Equation (1) is the one that provides the best fit to the mean points of the HFD group, and Equation (2) provides the best fit to the remaining three groups. Five curves were generated for each group, one for each experiment, and from the values obtained, V_max_, K_m ATP_, and K_i ATP_ were calculated (Figure 11B–D, respectively). For V_max_, K_m ATP_, and K_i ATP_, no significant differences were found between the CTR and HFD groups, nor when analyzing velocities with [ATP] = 5 mM (Student’s *t* = 0.66; *p* = 0.5269), a concentration close to that found in cardiomyocytes in both contraction and relaxation [11]. The differences in velocities between the two groups become significant at lower ATP concentrations, such as 1 mM (*t* = 5.65; *p* = 0.0005). Ang-(3–4) stimulates SERCA2a activity, increasing V_max_ and K_m ATP_ in both the CTR and HFD groups, decreasing K_i ATP_ in CTR rats, and causing the appearance of K_i ATP_ in HFD rats. However, the response of HFD animals concerning V_max_ and K_m ATP_ was lower than that of CTR animals, (Figure 11B–D).

### 2.6. There Is No Structural Remodeling in HFD Rats at a Juvenile Age

Figure 12 presents the results of echocardiographic studies designed to investigate whether there are indications of structural remodeling of the left ventricle in HFD rats. Measurements of the internal diameters of the left ventricle at the end of diastole and systole, as well as of the left-ventricular end-diastolic and end-systolic volumes, were recorded at the level of the region where the papillary muscles are located (red lines and circles). These measurements (representative recordings in panels A, B, C, and D, respectively) demonstrate that HFD rats do not present changes in the ejection fraction (Figure 12E) or the fractional shortening (Figure 12F), and that there were no effects of Ang-(3–4). Table 1 shows that there are also no differences in heart and left-ventricular weight among the four groups.

## 3. Discussion

The main and novel findings of the present work are that moderate weight gain (Figure 1) in young rats [12] chronically fed a high-fat diet [13] (HFD) is accompanied by a gradual development of arterial hypertension (Figure 2) and a marked and puzzling decrease in plasma triglyceride levels that appears to be associated with the steatosis that these rats present [6], together with decreased glucose tolerance (Figure 8). Hypertension in HFD rats is prevented by Ang-(3–4) administered at day 104 [6] (Figure 2). The rats presented increased visceral (epididymal and perirenal) fat (Figure 5). The left-ventricular levels of pro-inflammatory cytokines IL-6 and TNF-α increased (Figure 6), and the (Na^+^ + K^+^) ATPase and the ouabain-resistant Na^+^-ATPase decreased and increased, respectively, and the Ang II receptors’ abundance remained unmodified in the same preparation of left-ventricular membranes (Figure 10). The activity of SERCA2a loses its main regulatory property: the inhibition by excess of substrate (ATP) (Figure 11) also colocalized in the same microsome preparation in which the pro-inflammatory cytokines are upregulated.

Even though the abundance of Ang II receptors is similar in CTR and HFD rats, evidence of the role of upregulation of the RAAS axis coupled to Ang II type 1 receptors (AT_1_R) in the left ventricle arises from the observation that increased levels of the pro-inflammatory cytokines IL-6 and TNF-α in HFD rats return to normal values upon administration of Ang-(3–4). The participation of RAAS also emerges from the observation that, in the myocardium of HFD rats, the decrease in (Na^+^ + K^+^)ATPase and the increase in the ouabain-resistant Na^+^-ATPase are also normalized by Ang-(3–4), as well as the dependence on ATP concentration characterized by inhibition by excess substrate of SERCA2a. However, the important inflammatory and metabolic alterations and notable changes in ion transport ATPases from overweight rats still coexist with fractional shortening and ejection fraction unchanged.

Although moderate, the overweight of HFD rats is accompanied by a significant increase in visceral fat around the epididymis and in the perirenal region, an observation that also suggests hyperactivity of the local RAAS coupled with AT_1_R with downstream upregulation of the Ang II → AT_1_R axis (e.g., PKC and ERK [14]). It could be proposed—in terms of mechanisms—that the abnormal activation of this axis would promote lipogenesis in HFD rats [15,16,17].

The upregulation of IL-6 and TNF-α—which was reversed by Ang-(3–4)—in the same microsomal preparation in which Na^+^-transporting ATPases, SERCA2a and Ang II receptors are colocalized, is a predictor of future myocardial alterations. We chose to investigate IL-6 due to its known effect on establishing insulin resistance [18] and because it is considered a predictor of mortality in cardiorenal syndromes [19]. The upregulation of TNF-α levels in the left ventricle, in turn, reveals the existence of a link between a process of local inflammation caused by Ang II and subsequent fibrosis [20] and congestive heart failure [21]. In this context, the significant increase in IL-6 in CTR rats caused by Ang-(3–4) deserves special discussion in the context of the dual effects of this cytokine [22,23]. In normonourished animals, Ang-(3–4) would increase the release of the cytokine to exert anti-inflammatory effects, promoting the alternative activation of macrophages to the M2 phenotype [24].

A metabolic syndrome in HFD rats is evidenced by the lipidogram and blood glucose profiles indicating insulin resistance (Figure 7 and Figure 8). The biochemical remodeling of plasma lipoproteins with a decrease in HDL and an increase in LDL is classically described for the dyslipidemias of metabolic syndrome, which correlates with the increase in visceral fat discussed above. The maintenance of total cholesterol in the HFD group at CTR levels probably results from the combination of changes in lipid metabolism in different tissues that compensate for each other and reverberate in the plasma [25]. The marked and atypical decrease in plasma triglycerides could be linked to the pronounced hepatic steatosis that ultrasound reveals in HFD rats [6].

The components of metabolic syndrome include fasting hyperglycemia and glucose intolerance, completing a network of processes and mechanisms that associate visceral adiposity, insulin resistance, hypertension, changes in body Na^+^, and cardiovascular risk [2,26]. Although fasting hyperglycemia and insulin resistance are mild in the HFD group, it should be noted that these are young rats. Juvenile obesity, in addition to leading to a progressive worsening of insulin resistance over time, significantly increases cardiovascular risk in adulthood [27].

The inflammatory state of the myocardial tissue that the cytokines revealed and the incipient fibrosis process [6] indicate RAAS hyperactivity. This hyperactivity would result from the selective increase/decrease of targets in steps after the binding of Ang II to AT_1_R (and a decrease in signaling coupled to AT_2_R), because the abundance of both receptors in the left-ventricle microsomal fraction remained similar in the 4 groups as shown in Figure 9. Well-known targets downstream of Ang II binding to AT_1_R are MAP kinases (ERK1/2, JNK, p38, MAPK), NADPH oxidase (which would lead to increased production of reactive O_2_ species) [28,29], protein kinase C (PKC) (upregulated), and cyclic AMP-dependent protein kinase (PKA) (downregulated) [14,30]. These altered Ang II → AT_1_R and/or Ang II → AT_2_R signaling could very well be present in HFD rats.

The results obtained from Ang II receptors in cardiomyocytes contrast with what is found in renal proximal tubules. In the latter, there is a marked upregulation of AT_1_R suppressed by Ang-(3–4), concomitantly with the downregulation of AT_2_R. This contrast suggests that the primary lesion of the recently described hepatocardiorenal syndrome in overweight rats [6] may occur in the kidneys.

The impact of overweight on Na^+^-transporting ATPases (Figure 10) and SERCA2a (Figure 11) also constitutes a central element in the pathophysiology of cardiac structural and functional changes that can occur in HFD rats. Na^+^-transporting ATPases are relevant because they participate in events related to electrical activity and are involved in dysfunctions described in cardiometabolic diseases [31]. The downregulation of (Na^+^ + K^+^)ATPase potentiates apoptosis in cardiomyocytes [32] and oxidative damage [33], which would be especially critical when ROS production is increased by activation of the Ang II → AT_1_R axis [34]. Regarding the ouabain-resistant Na^+^-ATPase, its functioning in normonourished rats contributes to normal electrical activity [14]. The upregulation of this Na^+^ pump due to RAAS hyperactivity in HFD rats could lead to changes in the regulation of cardiomyocyte volume [35] and, therefore, in the function of these cells as a whole [36].

The relevance of potential changes in SERCA2a in HFD rats emerges from the fact that its changes impact the contractile process in heart failure [37]. As seen in Figure 11, there are no differences in SERCA2a activity at 5 mM ATP, a concentration close to that encountered in cardiomyocytes in both contraction and relaxation [11]. The fact that there is a difference in activity at low concentrations of ATP makes it evident that if myocardial inflammatory damage [38] in HFD animals is accentuated to the point of compromising mitochondrial function [39], the activity of SERCA2a will also be affected. A SERCA2a dysfunction that leads to decreased activity would accelerate the progression to heart failure [40,41] in HFD rats. Concerning the stimulation by Ang-(3–4), the fact that the response in HFD rats was smaller than in CTR rats suggests that chronic administration of the high-fat diet affected the ability of the Ang II → AT_2_R axis to counter-regulate the Ang II → AT_1_R axis. At this point, it is important to discuss the recovery of the kinetics of SERCA2a inhibition by excess substrate, one of the sophisticated mechanisms of key enzymes activity [42,43]. Inhibition of SERCA2a activity by high ATP makes it sensitive to energy demand during systolic sarcomere contraction during [44]. Thus, its loss in HFD rats, recovered by the administration of Ang-(3–4), again indicates a pathological effect at the molecular level resulting from the upregulation of the Ang II → AT_1_R axis of the RAAS.

Although the relationship between obesity and heart failure has been the subject of an increasing number of studies in recent years (for an illustrative review, see [45]), two questions have not yet been answered. They are: (1) Can moderate overweight be associated with or precede early heart failure? (2) What are the effects of inflammation and changes in ion transport in myocardial tissue associated with overweight/obesity? Cardiometabolic diseases in obesity, in which there is an important component of generalized inflammation, evolve and worsen when there is hypertension [46,47]. HFD rats became progressively hypertensive throughout the administration of the high-fat diet (Figure 2) and presented myocardial fibrosis [6]. The simultaneous renal and hepatic changes [6], with the intense steatosis that characterizes the latter and the RAAS-mediated upregulation of renal transport ATPases—leading to Na^+^ imbalance in body Na^+^ homeostasis—[2] also contribute to the molecular mechanisms that generate the hepatocardiorenal syndrome phenotype in HFD rats. This ensemble of observations adds to the changes in renal Ang II receptors already described [2]. They allow us to propose that the triad known as cardiovascular-kidney-metabolic syndrome [48] should even include moderate weight gains, which, as in the case of HFD rats, already have a significant expansion of visceral adiposity. In this initial stage of the evolution of fat mass and cellular and molecular changes in the myocardium, including marked inflammation and important changes in ionic transporters involved in excitation/contraction coupling, there is still no evidence of systolic dysfunction. As shown in Figure 12 and Table 1, both shortening fraction and ejection fraction, which are central measures of left ventricular systolic function [49,50,51], are preserved, as well as heart and left ventricular weight. Ang-(3–4) was also without effect, probably because the downstream upregulation of the Ang II → AT_1_R axis, discussed above, had not yet reached the macroscopic structural remodeling that echocardiography can reveal.

## 4. Materials and Methods

### 4.1. Ethical Considerations

All experimental procedures were approved by the Ethics Committee for Using Animals in Research (CEUA) at the Federal University of Rio de Janeiro under protocol number 075/19 (approved in 26 July 2019). The execution of these procedures adhered to the Committee’s Guidelines, which follow the Uniform Requirements for Manuscripts Submitted to Biomedical Journals established by the International Committee of Medical Journal Editors and the ARRIVE guidelines [52].

### 4.2. Experimental Groups, Diets, and Systolic Blood-Pressure Measurements

At birth, the rats (Wistar Hannover) were distributed so that each mother remained with 8 animals, ensuring that all had equal nutritional access [53]. All male animals were preserved, and, if necessary, the number of offspring per mother was raised to 8 by adding female pups. The animals were weaned on day 21 of life, when started to receive a commercial diet for rodents (Neovia Animal Nutrition, Contagem, Brazil), and were regrouped to avoid the litter effect [54] in cages containing 5 animals, where they remained until 57 days of life in a controlled environment at 23 ± 2 °C with a 12 h/12 h light and dark cycle.

Upon reaching 58 days of life, the offspring were randomly separated into 2 groups. The control (CTR), with free access to a diet containing 36, 292, and 57 kcal/100 g dry weight chow from lipids, carbohydrates, and proteins, respectively, totaling 385 kcal/100 g. The high-fat diet (HFD) group received 402, 103, and 69 kcal/100 g from lipids, carbohydrates, and proteins, respectively [13]. The Na^+^ content, measured by flame photometry, varied between 5.1–5.8 and 7.7–8.0 mequiv/100 g dry chow (CTR and HFD, respectively) [2]. For greater detail on the diets, see Supplement Table S1 in [2]. Both groups had free access to filtered water. Both CTR and HFD chow were purchased from PragSoluções (Jaú, Brazil). Upon reaching 104 days of diet (162 days of age), the rats were separated into 2 more groups—also randomly—and housed individually in metabolic cages so that half received by gavage 4 doses (each dose 80 mg/kg body mass diluted in water, 2 doses each day) of Ang-(3–4) (AminoTech, Sorocaba, Brazil). Untreated rats received only the vehicle. The individual dose of 80 mg/kg body mass was established from 2 classes of previous experiments. First, the dependence on the Ang-(3–4) concentration during *in-vitro* experiments on the inhibitory effects of ouabain-resistant Na^+^-ATPase in proximal tubules. Second, from in-vivo experiments in spontaneously hypertensive rats (SHR), studying the anti-hypertensive action of Ang-(3–4) and its influence on urinary volume, urinary Na^+^ concentration, and urinary Na^+^ excretion in 24 h, assuming that 1 kg of rat contains 0.7 L of water [3]. The fact that systolic blood pressure of SHR returned to high values 24 h after the administration of a single dose of 80 mg/kg Ang-(3–4) [3] led us to use 4 doses every 12 h in the present study, aiming to maintain the effect while the rats were housed in metabolic cages. The *n* values for body weight measurements were 30 and 26 for CTR and HFD rats, respectively.

After 48 h of treatment with Ang-(3–4), the rats were sacrificed by decapitation. Each animal’s heart, perirenal, and epididymal fat were immediately collected using surgical scissors. The organs were dried with filter paper to remove excess blood and weighed. Perirenal and epididymal fat masses were considered markers of visceral fat [55], as described above. The *n* values for epididymal fat measurements were 17, 13, 11 and 14 for CTR, CTR +Ang-(3–4), HFD, and HFD + Ang-(3–4) rats, respectively. The *n* values for perirenal fat measurements were 16, 13, 12 and 14 for CTR, CTR + Ang-(3–4), HFD, and HFD + Ang-(3–4) rats, respectively.

Systolic blood pressure was measured weekly, as indicated in Figure 2, using a tail-cuff plethysmograph V3.0 (Insight, Ribeirão Preto, Brazil). The rats were previously acclimated to the chamber several days before the recordings. The *n* values for systolic blood pressure measurements were 30 and 26 for CTR and HFD rats, respectively.

### 4.3. Lipidogram

After 70 days of consumption of the CTR or HFD diets (128 days of age), some of the rats were fasted for 10 h (water ad libitum) and anesthetized after this period with ethyl ether. The aim was to investigate whether the onset of hypertension at this age [2] is associated with alterations in lipid metabolism. Blood was collected by caudal puncture, centrifuged at 13,000× *g* for 5 min, and plasma was collected to measure total cholesterol/TC, HDL cholesterol, and triglycerides/TG. Commercial kits (Bioclin, Belo Horizonte, Brazil) were used: Bioclin K083 mono reagent total cholesterol, Bioclin enzymatic HDL cholesterol, and Bioclin K117 mono reagent triglycerides. In CTR and HFD groups, the *n* value was 10 rats.

### 4.4. Glycemia Determination and Glucose Tolerance Curve

With the same objective mentioned in the previous subsection, after 95 days of diet (153 days of age), some of the animals that received the CTR or HFD diets were fasted for 10 h (water ad libitum) to measure possible alterations in blood glucose. Immediately afterward, they received 2 g of glucose/kg body mass by gavage. The blood glucose measurement was done using the touch-ultra mini glucometer (OneTouch^®^-LifeScan, Milpitas, CA, USA), taking the fasting blood glucose value of time zero to start the curve. The measurements were taken 15, 30, 50, 60, and 120 min apart. The measurements of glucose and lipids were carried out at different times because, at 70 days, no differences were found between the plasma glucose concentrations of the CTR and HFD groups. In CTR and HFD groups, the *n* value was 10 rats.

### 4.5. Preparation of Left-Ventricle Microsomes

After sacrifice (at 106 days of exposure to CTR or HFD diets, 164 days of age), the heart of each animal was immersed in a solution containing 250 mM sucrose, 1 mM imidazole, 1 mM EDTA, and 0.15 mg/mL of trypsin inhibitor (pH adjusted to 7.6 with TRIS) [56]. Each heart was carefully dissected to obtain the left ventricle, weighed, and cut into smaller fragments with the aid of surgical scissors. A pool of 2–3 ventricles was suspended in the same solution (1 g of tissue in 4 mL).

The suspension was mechanically homogenized at 4 °C using a Potter Elvejhem homogenizer (RW 20 model, IKA Werke, Staufen, Germany) with a Teflon pestle (5 cycles of 1 min at 1700 rpm with 10 s pauses). Homogenates were sequentially centrifuged twice (1650× *g* for 15 min at 4 °C, 70 Ti Beckman rotor) to remove debris and nuclei, and then the supernatant was centrifuged at 115,000× *g* for 60 min. The resulting precipitate was recovered and suspended in 250 mM sucrose solution and stored in liquid N_2_. Protein concentration was determined by the Lowry method [57]. This microsomal fraction has a sarco-endoplasmic reticulum content of 8–10%, which allows simultaneous assessment of (Na^+^ + K^+^)ATPase, ouabain-resistant Na^+^-ATPase, and sarco-endoplasmic reticulum Ca^2+^-ATPase 2a (SERCA2a) activities.

### 4.6. SDS PAGE and Western Blotting

Western blotting evaluated the abundance of (Na^+^ + K^+^)ATPase, Ang II receptors, IL-6, and TNF-α. Microsome samples (80 μg of protein) were loaded onto a gel containing 10% polyacrylamide and SDS [58]. After the separation and transfer to nitrocellulose membranes (GE Healthcare, Chicago, IL, USA), the membranes were then incubated for 1 h in a solution containing 5% skimmed milk powder diluted in Tris-buffered saline (pH 7.6). The membranes were incubated overnight with the corresponding primary antibody, then incubated during 1 h with the secondary antibody for subsequent development.

The primary antibodies were as follows. Against the α-catalytic subunit of (Na^+^ + K^+^)ATPase, the catalog antibody A276 (Sigma-Aldrich, Saint Louis, MO, USA) was used at a dilution of 1:1000. For the cytokines IL-6 and TNF-α, antibodies from Imuny Biotechnology (Campinas, Brazil; catalog IM-0407) at a dilution of 1:1000 and from Imuny Biotechnology (catalog IM-406) at a dilution of 1:1000 were used, respectively. For AT_1_R, the anti-rabbit polyclonal antibody was used (Alomone Labs, Jerusalem, Israel; catalog AAR-011, at a dilution of 1:500). For AT_2_R, the anti-rabbit polyclonal antibody (Santa Cruz Biotechnology, Dallas, TX, USA; catalog SC-9040, dilution 1:250). The secondary antibodies were anti-mouse and anti-rabbit from GE Healthcare (NA934 and NA931, respectively), both at 1:5,000 dilution. The loading control was performed through immunodetection of GAPDH (monoclonal antibody SC-32233, Santa Cruz Biotechnology). Since stripping a membrane removes primary and secondary antibodies without affecting membrane proteins, we assume that GAPDH is unchanged. Therefore, probing of GAPDH was performed last, and only once, after sequentially probing IL-6 and TNF-α (Figure 6) or AT_1_R and AT_2_R (Figure 9). Since samples have unequal protein loading, we constantly analyze GAPDH densitometry (two-way ANOVA). No differences were found within each group of membranes (see Appendix A). The LAS 4000 system equipment (GE Healthcare) and the ImageJ software (NIH) were used to detect the immunosignals. For AT_1_R receptors, the *n* value was 8 in all groups. For AT_2_R receptors, the *n* values were 8, 8, 8 and 7 microsomal preparations (each preparation corresponds to 1 rat) in the CTR, CTR + Ang-(3–4), HFD, and HFD + Ang-(3–4), respectively. For (Na^+^ + K^+^)ATPase, the *n* value was 4 in all groups. For IL-6 and TNF-α, the *n* value was 6 in all groups.

### 4.7. Measurement of Left-Ventricle Na^+^-Transporting ATPase Activities

Activities of the two Na^+^-transporting ATPases were assessed by measuring the orthophosphate released during ATP hydrolysis [59]. For the (Na^+^ + K^+^)ATPase experiments, microsome samples (0.05 mg/mL) were preincubated at 37 °C for 10 min in a solution containing 50 mM Bis-TRIS propane (pH 7.4), 0.2 mM EDTA, 120 mM NaCl, and 5 mM MgCl_2_, in the absence or presence of 5 mM ouabain. The reaction was started by adding a solution containing 5 mM ATP and 24 mM KCl (final concentrations) and terminated 10 min later by adding a suspension of activated charcoal in 1 N HCl (1 vol medium:1 vol charcoal suspension). For the subsequent colorimetric reaction, the reaction tubes were centrifuged at 13,000× *g* for 15 min to remove charcoal. Next, the supernatant was removed and mixed with ferrous sulfate-molybdate solution (1 vol supernatant:1 vol ferrous sulfate-molybdate solution). (Na^+^ + K^+^)ATPase activity was calculated by subtracting the activity obtained in the presence of ouabain from the total activity.

The ouabain-resistant and furosemide-sensitive Na^+^-ATPase activity was assessed by first preincubating microsome samples (0.2 mg/mL) in a solution containing 20 mM Hepes-TRIS (pH 7.0), 10 mM MgCl_2_, 120 mM NaCl, and 2 mM ouabain in the presence or absence of 2 mM furosemide. The reaction (at 37 °C) was started by adding 5 mM ATP and finished 10 min later, following the process described for the (Na^+^ + K^+^)ATPase. The activity of the ouabain-resistant and furosemide-sensitive Na^+^-ATPase was determined by the difference between the activities measured in the absence and presence of furosemide. For (Na^+^ + K^+^)ATPase and Na^+^-ATPase, the *n* value was 8 in all groups.

### 4.8. Measurement of Left-Ventricle Sarco-Endoplasmic Reticulum Ca^2+^-ATPase/SERCA2a Activity

SERCA2a activity was assayed at increasing ATP concentrations (0.1, 0.2, 0.5, 1, 3 and 5 mM). Microsomes (0.05 mg/mL) were initially preincubated at room temperature for 20 min in the absence or presence of 4.6 μM thapsigargin, in a solution containing 50 mM TRIS-HCl (pH 7.0), 160 mM sucrose, 10 mM NaN_3_, 0.2 mM ouabain, 1.0 mM EGTA, and sufficient MgCl_2_ and CaCl_2_ concentrations [60] to obtain free Mg^2+^ and Ca^2+^ concentrations of 0.5 mM and 20 μM, respectively, for each ATP concentration. After preincubation, ATP was added to obtain the final concentrations mentioned above. After incubation for 15 min at 37 °C, the reaction was stopped by adding the charcoal suspension in acid and the samples processed as in the case of Na^+^-transporting ATPases. SERCA2a activity was calculated as the difference between total activity and that obtained in the presence of thapsigargin. For V_max_ and K_m ATP_ determinations, the *n* value was 5 in all groups. For K_i ATP_, the *n* values were 4, 5, and 5 microsomal preparations in the CTR, CTR + Ang-(3–4), and HFD + Ang-(3–4), respectively.

### 4.9. Echocardiographic Images

Echocardiography studies were performed to evaluate cardiac structure/function in vivo. The studies were conducted using a Vevo 2100 High-Resolution Imaging System (VisualSonics, Toronto, ON, Canada) coupled to a 21 MHz transductor. Rats (aged 164 days; 106 days of exposure to the different diets, including the two days under Ang-(3–4 treatment) were maintained under isoflurane anesthesia during the study. Images were acquired in bidimensional and M mode [61] and then analyzed by a blinded specialist using the VevoLab software (version 5.6.1). The *n* values were 17, 13, 12, and 14 rats for CTR, CTR + Ang-(3–4), HFD, and HFD + Ang-(3–4) groups, respectively (fractional shortening and ejection fraction).

### 4.10. Statistical Analysis

Results, which were analyzed using GraphPad Prism 8.0.2, are presented as mean ± SD. Normal distribution was checked using the Shapiro-Wilks test. Two means were compared using the paired or unpaired Student’s *t*-test. For comparisons between more than two means, two-way ANOVA was used (diet and Ang-(3–4) as factors). One-way ANOVA was used to compare the evolution of systolic blood pressure in CTR and HFD rats that did not receive (Ang-(3–4). In the glucose tolerance analysis, Student’s *t*-test was employed, according to [9]. The number of rats in each group differed because the data obtained with rats that died on day 106 during anesthesia with isoflurane in the echocardiographic studies were not used. Differences were considered significant at *p* < 0.05.

## 5. Conclusions

In conclusion, our research demonstrates that even when rats are only very moderately overweight there are critical processes of molecular damage in myocardial tissue, which are revealed by inflammatory biomarkers. Even though the fractional shortening and ejection fraction are preserved, they may be altered if the inflammatory process persists. The results described in the present study elucidate the molecular mechanisms underlying the biological processes involved. The accumulation of visceral fat, producing pro-inflammatory cytokines that reach and accumulate in the myocardium, leads to fibrosis. The cellular environment of inflammation and myocardial fibrosis impacts different molecular targets, notably the kinetics of SERCA2a and its regulation by the RAAS, which anticipates structural and functional remodeling of the left ventricle. The downregulation of (Na^+^ + K^+^)ATPase and the upregulation of ouabain-resistant Na^+^-ATPase are responsible for ionic changes distribution able to impact the coupling excitation:contraction.

## 6. Future Directions

Future directions of the study include the challenge of characterizing altered components of signaling pathways downstream of the cardiac Ang II receptors, whose abundance remains unmodified in moderate overweight. At this initial pathological stage, it will be important to detect intervention targets that can prevent progression to systolic dysfunction and cardiac remodeling.

## Figures and Tables

**Figure 1 ijms-25-12474-f001:**
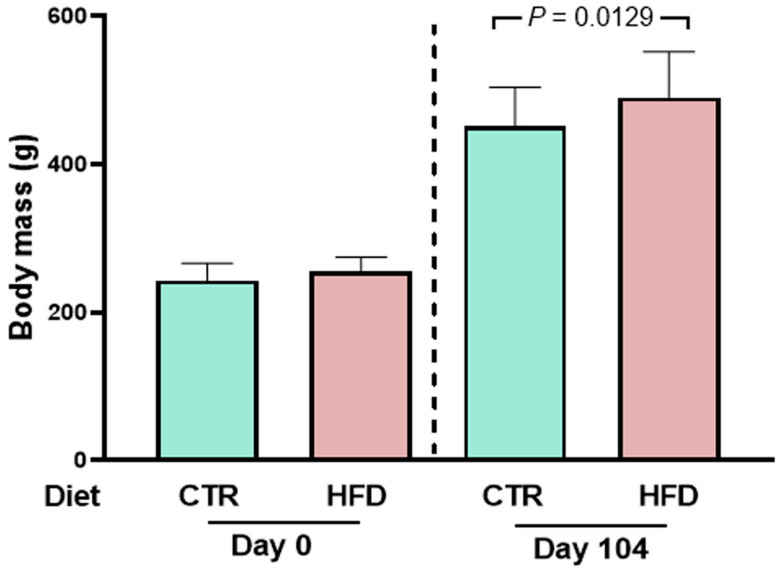
Administration of the high-fat diet (HFD) over 104 days starting at 58 days of age leads to a small but significant overweight of male rats. Left panel: body mass at the beginning of different dietary exposures (zero time). Right panel: body mass 104 days later. Diets and days of exposure to the different diets are indicated on the abscissae; *n* = 30 (CTR rats) and *n* = 26 (HFD rats). Bars show mean ± SD. Differences between CTR and HFD were assessed on the same day using unpaired Student’s *t*-test; *p* value (Day 104) is indicated within the panel.

**Figure 2 ijms-25-12474-f002:**
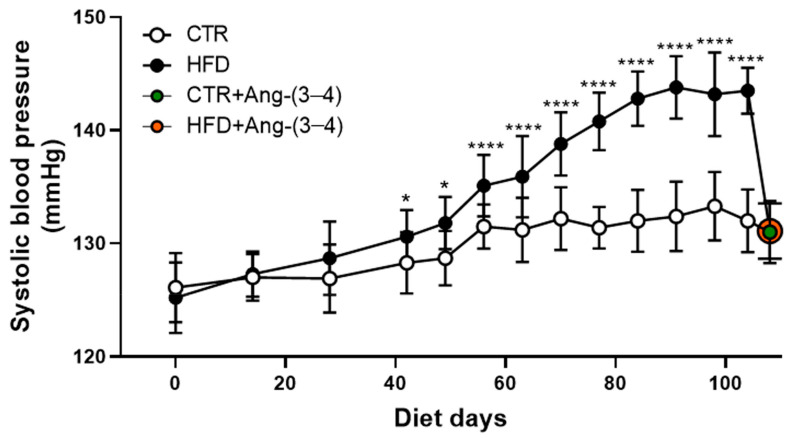
Evolution of systolic blood pressure in CTR (*n* = 30) and HFD (*n* = 26) rats. The data points show mean ± SD. Differences were assessed using one-way-ANOVA followed by Bonferroni’s test for selected pairs (CTR and HFD rats of the same age). * *p* < 0.05; **** *p* < 0.0001. The Ang-(3–4) data at day 106 (green circle in the CTR + Ang-(3–4) group; orange circle in HFD + Ang-(3–4) rats) are reproduced from Crisóstomo et al. [6] under the terms of Creative Common License CC BY 4.0.

**Figure 3 ijms-25-12474-f003:**
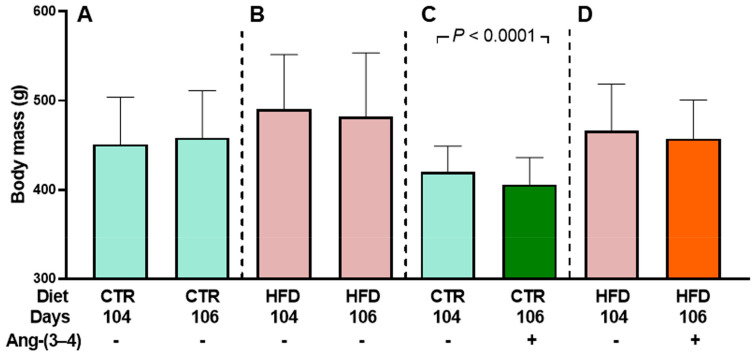
Evolution of body mass among rats housed in metabolic cages: Effect of Ang-(3–4) administration [4 doses (2 doses per day) on days 104 and 105]. Diets, days of exposure to the different diets at the end of the study, and administration or not of Ang-(3–4) are indicated on the abscissae. CTR (*n* = 30) and HFD (*n* = 26) rats were divided into two subgroups. One of them was used to follow body mass evolution without Ang-(3–4) treatment between days 104 and 106 (*n* = 17 and 12 for CTR and HFD, respectively) (**A**,**B**). The other subgroup was used to follow body mass evolution after Ang-(3–4) administration (*n* = 13 and 14 for CTR and HFD, respectively) (**C**,**D**). Bars show mean ± SD. Differences between days 104 and 106 in each subgroup were assessed using paired Student’s *t*-test; *p* values are indicated within the panels. Even though the untreated CTR body mass values at 104 days in (**A**,**C**) are numerically different after the random formation of the subgroups, and the same applies for untreated HFD rats at 104 days in (**B**,**D**), there were no statistical differences within the pairs of CTR and HFD groups (*p* = 0.0569 and 0.2285, respectively; unpaired Student’s *t*-test).

**Figure 4 ijms-25-12474-f004:**
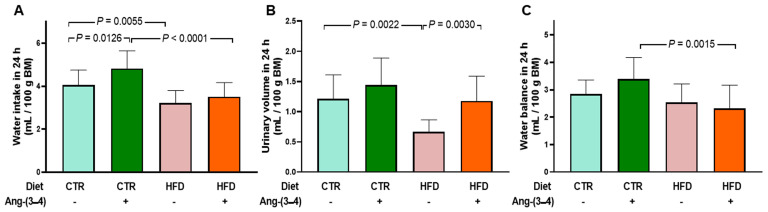
Water intake (**A**), urinary volume (**B**), and water balance (**C**) in 24 h measured on day 106. Measurements were carried out after administering or not 2 doses of Ang-(3–4) on days 104 and 105 (total of 4 doses). Diets and administration or not of Ang-(3–4) are indicated on the abscissae; *n* = 10–16. Bars show mean ± SD. Differences were assessed using two-way ANOVA followed by Bonferroni’s test; *p* values are indicated within the panels.

**Figure 5 ijms-25-12474-f005:**
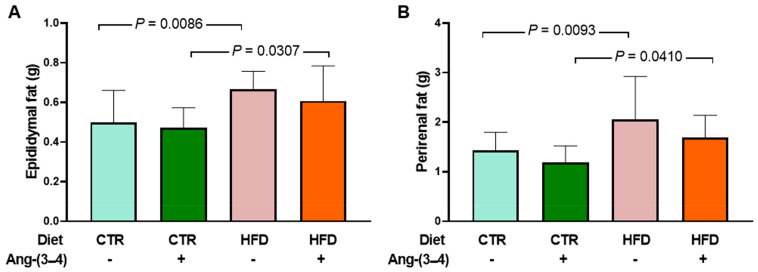
Augmented visceral fat in HFD rats. Measurements were carried out on day 106 after administering or not 4 doses of Ang-(3–4) (2 each day) on days 104 and 105. Diets and administration or not of Ang-(3–4) are indicated on the abscissae. (**A**) Epididymal fat. (**B**) Perirenal fat. Bars show mean ± SD (*n* = 11–17). Differences were assessed using two-way ANOVA followed by Bonferroni’s test; *p* values are indicated within the panels.

**Figure 6 ijms-25-12474-f006:**
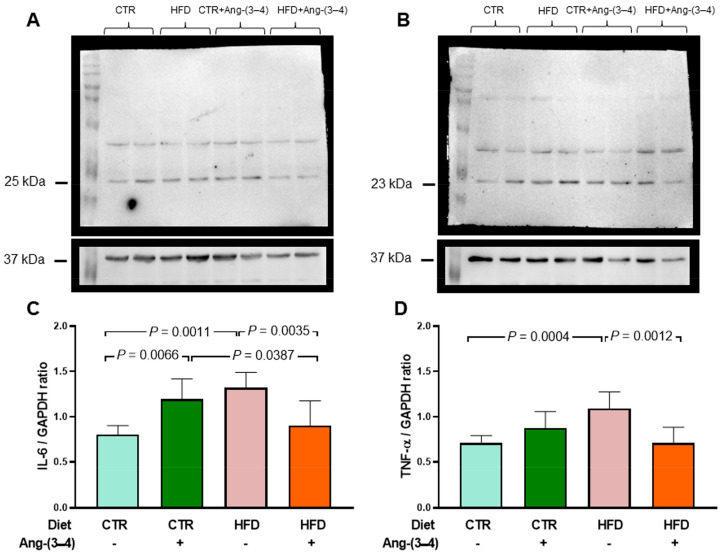
Increased proinflammatory cytokines in microsomes of left-ventricle cardiomyocytes from HFD rats: recovery of control levels after Ang-(3–4) administration. (**A**) Representative Western blots of IL-6 (25 kDa). (**B**) Representative Western blots of TNF-α (23 kDa). Upper lanes, cytokines; bottom lanes, loading control, GAPDH (37 kDa). (**C**) Quantification of IL-6 levels. (**D**) Quantification of TNF-α levels. After administering 4 doses of Ang-(3–4) on days 104 and 105 (2 each day), measurements were carried out on day 106. Diets and administration or not of Ang-(3–4) are indicated above the blots (**A**,**B**) and on the abscissae (**C**,**D**). Bars show mean ± SD (*n* = 6 different preparations of cardiac microsomes that were the same for measuring both cytokines). Differences were assessed using two-way ANOVA followed by Bonferroni’s test; *p* values are indicated within the panels.

**Figure 7 ijms-25-12474-f007:**
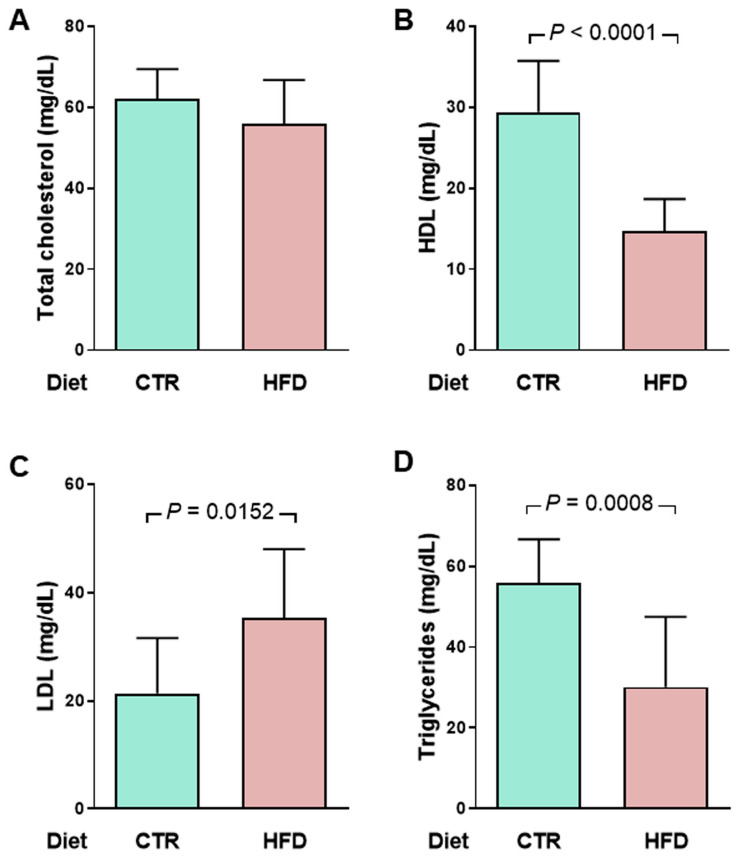
An early atypical lipidogram in overweight rats was encountered after 70 days of exposure to the HFD diet (128 days of life). (**A**) total cholesterol (TC), (**B**) high-density lipoproteins (HDL), (**C**) low-density lipoproteins (LDL), and (**D**) Triglycerides (TG) were measured in the same plasma samples (*n* = 10 for CTR and HFD groups). LDL cholesterol was estimated using the empirical formula LDL = TC − HDL − TG/5 [8]. Diets are indicated on the abscissae. Bars show mean ± SD. Differences were assessed using unpaired Student’s *t*-test; *p* values are indicated within the panels.

**Figure 8 ijms-25-12474-f008:**
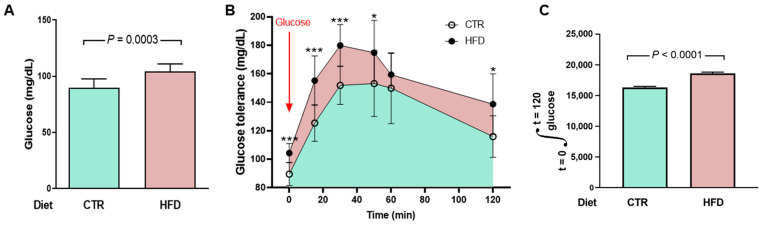
Moderate hyperglycemia and impaired glucose tolerance in HFD rats. (**A**) After 95 days on the different diets (153 days of life), the rats (*n* = 10 for CTR and HFD) were fasted for 12 h for a first determination of blood glucose (groups indicated on the abscissa). Bars correspond to mean ± SD. The difference was analyzed using unpaired Student’s *t*-test (*p* value shown within the panel). (**B**) After blood collection to determine fasting blood glucose, the rats received glucose by gavage (arrow), and plasma concentrations were measured at the times indicated on the abscissa. The points correspond to mean ± SD. Statistical comparison between means at each time point was performed using the unpaired Student’s *t*-test, following [9]. * *p* < 0.05, *** *p* < 0.001. The blue area shows the area under the curve in CTR rats and the pink area shows the extra area under the curve in HFD rats. (**C**) Quantifying the area under the glucose tolerance curves corresponding to CTR and HFD rats. Bars represent mean ± SD. The difference was investigated using the unpaired Student’s *t*-test (*p* value indicated within the panel).

**Figure 9 ijms-25-12474-f009:**
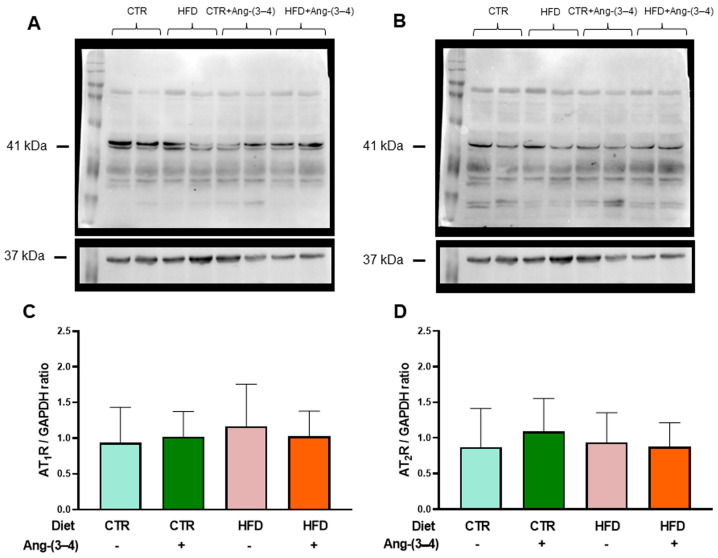
The abundance of Ang II receptors in microsomes of left ventricle cardiomyocytes in HFD rats remains unmodified at day 106. AT_1_R (**A**,**C**) protein levels and AT_2_R (**B**,**D**) were measured after 106 days of exposure to the different diets. Ang-(3–4) (4 doses; 2 doses each day) was administered during days 104 and 105. Combinations of diets and treatment or not with Ang-(3–4) are indicated above the blots in (**A**,**B**), where 41 and 37 kDa correspond to the molecular masses of Ang II receptors and the loading control GAPDH, respectively. The combinations are also indicated on the abscissae of the receptor’s quantification panels (**C**,**D**), which show mean ± SD; *n* = 7–8. Differences were assessed using two-way ANOVA.

**Figure 10 ijms-25-12474-f010:**
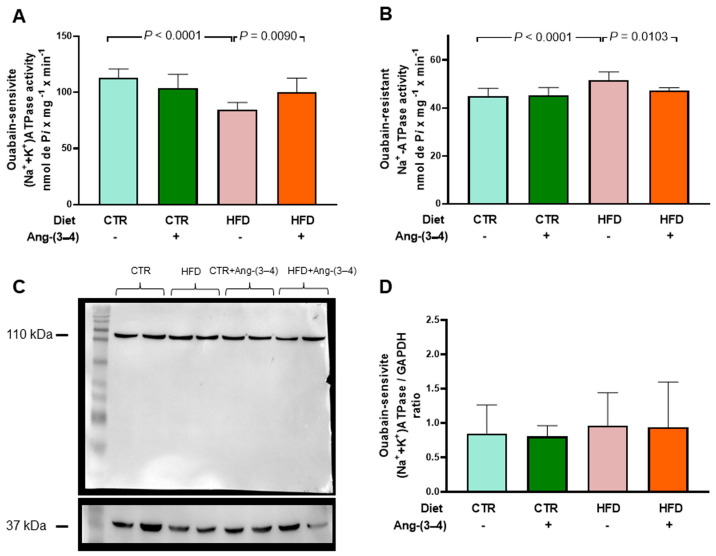
Opposite profiles of Na^+^-transporting ATPase activities in microsomes of left-ventricle cardiomyocytes from HFD rats. Combinations of diets and treatment or not with Ang-(3–4) are indicated on the abscissae and above the lanes of the representative Western blot. (**A**) Ouabain-sensitive (Na^+^ + K^+^)ATPase activity. (**B**) Ouabain-resistant, furosemide-sensitive Na^+^-ATPase activity. Bars show mean ± SD; *n* = 8 in all cases, using different microsome preparations. (**C**) Representative Western blot of ouabain-sensitive (Na^+^ + K^+^)ATPase; 110 kDa and 37 kDa correspond to the molecular masses of the α-catalytic subunit of (Na^+^ + K^+^)ATPase and GAPDH, respectively. (**D**) Quantification of the immunodetections of ouabain-sensitive (Na^+^ + K^+^)ATPase corrected by the loading control. Bars show mean ± SD; *n* = 4 microsome preparations. In (**A**,**B**,**D**), the differences were assessed using two-way ANOVA followed by Bonferroni’s test; *p* values are indicated within the panels.

**Figure 11 ijms-25-12474-f011:**
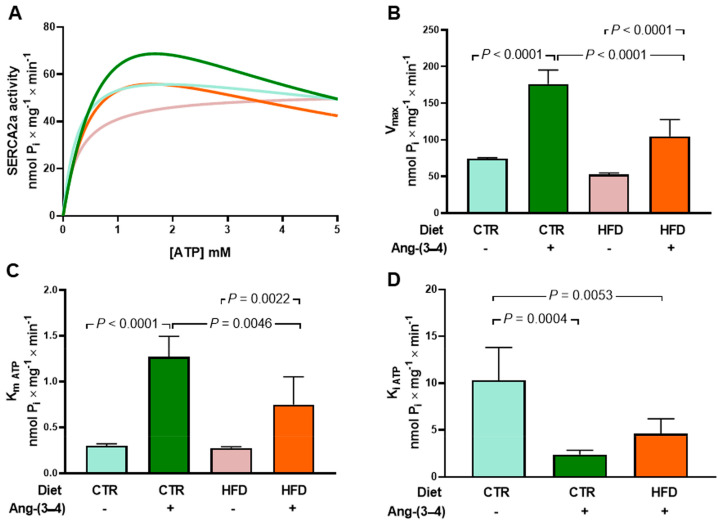
Abnormal kinetics of sarco-endoplasmic reticulum Ca^2+^-ATPase (SERCA2a) in microsomes of left-ventricle cardiomyocytes from HFD rats: partial recovery of the control ATP concentration dependence in Ang-(3–4)-treated HFD animals. (**A**) Simulations of ATP concentration dependence of SERCA2a between 0.1 and 5 mM using experimental mean values of enzyme velocity (*n* = 5 different microsome preparations). The curves were generated using equations 1 or 2 (see text). Blue, CTR; pink, HFD; green, CTR + Ang-(3–4); orange, HFD + (Ang-(3–4). (**B**) V_max_ values (nmol P_i_ × mg^−1^ × min^−1^). (**C**) K_m ATP_ (mM). (**D**) K_i ATP_ (mM). In (**B**,**C**,**D**), bars show mean ± SD (*n* = 4–5). Combinations of diets and treatment or not with Ang-(3–4) are indicated on the abscissae. Differences were assessed using two-way ANOVA followed by Bonferroni’s test (**B**,**C**) and one-way ANOVA followed Bonferroni’s test for selected pairs (**D**); *p* values are indicated within the panels.

**Figure 12 ijms-25-12474-f012:**
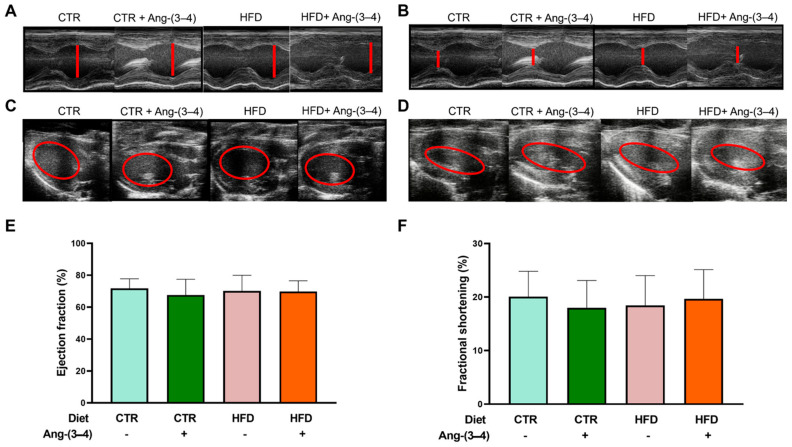
Echocardiography at day 106 reveals unchanged structural and functional parameters of the left ventricle behind the metabolic, cellular, and molecular changes found in overweight rats. Measurements of the left-ventricular internal diameter at the end of diastole (LVIDd) and the end of systole (LVIDs) allow calculation of the fractional shortening (FS). Measurements of the left-ventricular end-diastolic volume (EDV) and end-systolic volume (ESV) allow calculation of ejection fraction (EF). Representative echocardiographic recordings of LVIDd (**A**) and LVIDs (**B**). Representative echocardiographic recordings of EDV (**C**) and ESV (**D**). (**E**) Ejection fraction calculated as EF (%) = [(EDV − ESV)/EDV] × 100. (**F**) Fractional shortening calculated as FS (%) = [(LVIDd − LVDIs)/LVIDd] × 100. Bars represent mean ± SD, which were compared using two-way ANOVA followed by Bonferroni’s test (*n* = 12–17 rats). Combinations of diets and treatment or not with Ang-(3–4) are indicated above the images and on the abscissae of the graphic representations.

**Table 1 ijms-25-12474-t001:** Heart and left-ventricular weight in CTR and HFD rats, treated or not with Ang-(3–4).

Total Heart Weight (g)	*n*	Mean	SD
CTR	17	1.41	0.19
CTR + Ang-(3–4)	13	1.29	0.16
HFD	12	1.45	0.16
HFD + Ang-(3–4)	14	1.40	0.11
Left-ventricular weight (g)	*n*	Mean	SD
CTR	17	1.15	0.17
CTR + Ang-(3–4)	13	0.99	0.13
HFD	12	1.14	0.23
HFD + Ang-(3–4)	14	1.09	0.18

## Data Availability

The data presented in this study are available on request from the corresponding author.

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
