# Peer review of "Male Wistar Rats Chronically Fed with a High-Fat Diet Develop Inflammatory and Ionic Transport Angiotensin-(3–4)-Sensitive Myocardial Lesions but Preserve Echocardiographic Parameters"

_ijms, 2024, doi:10.3390/ijms252212474_

Round 1

Reviewer 1 Report

Comments and Suggestions for Authors

The manuscript investigates the cardiometabolic effects of a high-fat diet on young male rats, particularly focusing on the role of Ang-(3–4) in modulating these effects. The study provides insights into the molecular and cellular alterations in the myocardium that precede observable echocardiographic changes. The authors report significant findings related to inflammatory markers, ionic transporters, and the role of the renin-angiotensin-aldosterone system (RAAS) in the development of myocardial lesions.

1. The authors should enhance the context and structure of the manuscript to improve readability and ensure the narrative flows logically.

2. The rationale for selecting a single dosage of 80 mg/kg of Ang-(3-4) is unclear. The authors should consider conducting dose-response and time-response studies to identify the optimal dose and range of effective doses.

3. The study assessed inflammation by measuring IL-6 and TNF-alpha levels in LV microsomes. However, a more comprehensive evaluation of additional inflammation markers is recommended for a thorough assessment.

4. The authors need to clarify the observed decrease in body mass in Figure 2, specifically the discrepancies between CTR 104 in panel A vs CTR 104 in panel C, and HL 104 in panel B vs HL 104 in panel D. The explanation of these results requires improvement.

5. The results presented in Figure 4 should be supported with corresponding images to enhance the validity of the findings.

6. The use of the same GAPDH blot in Figures 5A, 5B, 8A, and 8B is problematic. The authors should include individual GAPDH blots after each stripping to ensure accurate loading control.

7. The authors need to justify the observed changes in GAPDH protein expression, as this suggests unequal protein loading among samples. It is advised to refine their technique and improve the quality of the blots for consistency.

8. While the authors conducted a lipidogram with the HL diet, it would be insightful to include lipidogram levels following Ang-(3-4) treatment to assess the compound's impact on lipid profiles.

9. The authors should elucidate the molecular mechanisms underlying their findings to better understand the biological processes involved.

10. The entire manuscript requires a thorough revision to enhance clarity, ensuring that the scientific content is presented in a clear and comprehensible manner.

Author Response

We prepared a unified response file, point-by-point (to the 6 Reviewers).

Reviewer 2 Report

Comments and Suggestions for Authors

In the current study the authors continued their previous study (ref. 27) and report about molecular adaptations in male rats. The authors show that moderate overweight causes cardiometabolic changes and disruption of Na and Ca transport.

Major comment:

It is hard to see the aim (central hypothesis) of this study and the conclusion. I.e., in the abstract the authors say that they “investigated inflammatory lesions and ion transporters in the left ventricle of male rats chronically fed a high-fat diet (HL)”. As the conclusion of their study however, the authors say that “even very modest overweight can cause silent initial damage to the heart that can evolve to systolic dysfunction.” However, what exactly is a silent initial damage and why is their systolic dysfunction in these rats? Moreover, what about the inflammation as this is the aim of the study?

Overall, I can hardly follow the arguments of the authors. There is no information about blood pressure but the authors talk about hypertension. There was a moderate increase in blood pressure reported in ref. 27 but are these the same rats?

The authors claim that they follow the ARRIVE guidelines. However, this is not the case. There is no information about the rat strain. Unequal group sizes without any explanation why different numbers of rats were used for each group is a problem. Obviously, the authors report about two-side ANOVA not two-way ANOVA. Please replace SEM by SD to give the reader an impression about the variation. How did the authors check normal variability and distribution of their samples?

If the weight of the heart and LV was determined, why did you not report these important data?

What does it mean: “These accentuated molecular lesions can alter the left 620 ventricle's hemodynamic function, which is still preserved.” Is it altered or not?

You write: “The main findings of the present work are that moderately overweight (Figure 1) in 342 young rats [13] chronically fed a high-fat and high-caloric diet [14] (HL) present – in ad- 343 dition to the gradually installed arterial hypertension – changes in central areas that reg- 344 ulate thirst [32] since liquid intake is markedly lower and, consequently, urinary excretion 345 is too (Figure 3).”

How can this be the main finding if it was not the aim of the study to investigate this relationship and this is not even mentioned in the abstract?

Author Response

(The authors gave the same response as above.)

Reviewer 3 Report

Comments and Suggestions for Authors

Authors present an interesting paper about the effects of high fat diet and Ang (3-4) on cardiac function. However, there are a number of major an minor issues.

Major.

1. English language. Quite a bit of problem with word usage and grammar. 

2. The abstract says "silent initial damage to the heart that can evolve to systolic dysfunction". However, that was not shown in the study itself, so it is better not to emphasize.

3. Terminology: it is better to settle on a more neutral and descriptive terms of the overfed rats like "high-fat-diet rats" or "HFD rats" for short. This way the question whether the rats are overweight or obese becomes non issue. The terms overweight and obese are clearly defined for humans, but not for animal models.

4. The Results  section is overloaded with rationale and discussion text. These passages needs to be either dropped or made more concise. 

5. The text describing results related to the Fig. 2 is missing.

6. line 153. "Ang 152 (3–4) significantly modifies visceral fat levels". This statement is not justified because both HL- vs HL+ are not significant.

7. Discussion is a bit too wordy and unfocused.

8. Conclusions are rather wordy with unnecessary phrases and passages like future directions. Please, state briefly what was found and what those finding mean.  

9. Authors need to highlight better what is novel in their results. 

For minor issue please see attached file.

Comments on the Quality of English Language

Main point: Overweight is an adjective and obesity is a noun. Obesity is a condition. Overweight is a weight status: underweight, healthy, overweight, obese.

Please review for word usage, grammar, and clarity.

Author Response

(The authors gave the same response as above.)

Reviewer 4 Report

Comments and Suggestions for Authors

This study contains a huge amount of data.
The data presented are very rich.
I believe that it needs explanations in many points:
1.     It is mentioned that hypertension occurred. Measurements are not explicitly             given.
2.     (35-321) Structural changes are mentioned but not documented
        (74-77) this statement is  confusing.
        Figure 2 is confusing
        (375-382) statements should be explained.  
       A lot of data are derived from Ref.27. This should be better described.
3.     (34) It is stated that changes in contractility occurred, which cause silent        initial damage to the heart that can evolve to systolic dysfunction
       This is not supported by the findings.

Author Response

(The authors gave the same response as above.)

Reviewer 5 Report

Comments and Suggestions for Authors

In the study the cardiometabolic, inflammatory and molecular changes in young and moderately overweight male rats receiving a high-fat, high-calorie diet have been investigated.

I have some questions and comments:

1.   Figure 1 shows that mean value of body mass of rats after administration of high-fat diet during 104 days was 506 g. After random formation of subgroups (Figure 2B and 2D) was in both subgroups mean value under 500 g. Could you comment it?

2.    Body mass of control animals on Day 106 was in animals treated with Ang-(1-3) significantly lower than in untreated rats. From this point is interesting finding that the amount of epididymal fat was in both group very similar. Another point is that animals in groups treated with Ang-(1-3) had after random selection on Day 104 lower body mass. This can influence the interpretation of some data obtained on Day 106.

3.  The protein levels of IL-6 and TNF-alpha you determined in microsomes. Microsomes are a good choice for the determination of transport ATPases activities but I think that data from microsomes do not have very informative value in terms of modulating the processes of inflammation.

Do you have data also about the levels of these proteins in soluble tissue extracts and in blood plasma/serum (released circulating proteins)?

4.    Why did you determine changes in lipid metabolism after 70 days of exposure to high-fat diet, glucose metabolism after 95 days on diet and other parameters 104 days after HF diet?

     Was lipid metabolism after HF diet influenced in comparison to control conditions after 104 days on diet? Did application of Ang-(1-3) influence the lipid profile?

5.   You mentioned several times that the rats on high-fat diet were hypertensive. Did you measure blood pressure during the present study? If yes, were changes in blood pressure influenced by Ang-(1-3)?

Author Response

(The authors gave the same response as above.)

Reviewer 6 Report

Comments and Suggestions for Authors

The study of Crisóstomo at al. investigated cardiometabolic changes in young moderately overweight hypertensive male rats receiving a high-fat, high-calory diet, with the aim of looking for possibly inflammatory cellular and molecular changes and some alteration in the population of ion transporters in the left ventricle, potentially associated with hyperactivity of the local cardiac RAAS arm coupled to type 1 Ang II receptors. In addition effect of Ang-(3-4) has been investigated. The study is potentially interesting, but I have a several questions and comments about it:

Major comment:

Statistical analysis of the results should be done using a 2-way factorial ANOVA with the factors "diet" and "Ang-(3-4) treatment" followed by suitable post-hoc test. This would enable the authors to find out also the main effects of these factors. If parameters were measured repeatedly in individual rats (as was the case for BW and the glucose tolerance test), ANOVA for repeated measures should be used. Comparison of groups by Student's test is not appropriate in this study and the statistical analysis should be corrected before publication.

 Another comments:

Line 26: In the Abstract, it is written  „IL-6 and TNF-α doubled as HL rats became moderately overweight...“. However, the two-fold increase does not follow from the results of the study. It is necessary to indicate more precisely the values ​​of the increase of these markers.

Line 27: Also, the statement „when signaling coupled to Angiotensin II type 2 receptors  (AT2R) was up-regulated by Ang-(3–4) administration“.....is not clear, it is necessary to state exactly which signal pathways have been activated.

Line 65: In the Introduction, it is written that hypertensive animals were used. Also in Discussion is written „The main findings .....– in addition to the gradually installed arterial hypertension-...“ (line 343). However, in the entire manuscript it is not written which rat strain was used and what was blood pressure of rats. Blood pressure data must be added to the manuscript or comments on blood pressure should be completely excluded from the Abstract, Introduction and the main findings of this study. They should be only discussed and cited, if the authors already published them in the another article.

Lines 74-78: These conclusions should not be in the Introduction but in the Conclusions. Instead, authors should write what hypotheses were tested in this study.

Results in general: The entire Results section is written rather narratively. I would appreciate more exact data, including accurate statistical analysis. All explanatory sequences should rather be in the Introduction or in the Discussion, where appropriate. The results should only contain data, figures and statistical analysis. For better clarity, the results of comparisons of statistically insignificant differences should not be in the figures, but rather in the text. In the figures, it is more appropriate to indicate only significant differences to improve their readability.

The use of GADPH as a housekeeper protein and loading control does not seem appropriate in this study. From the original (as well as representative 8A, 8B) membranes, it seems  that Ang-(3-4) can reduce the expression of GADPH in control conditions. The authors should add a the statistical analysis of GADPH expression (as Supplementary results) and show, using a 2-way factorial analysis ANOVA, that GADPH expression does not change under the experimental conditions used in this study, especially when Ang-(3-4) is administered.

Line 207: An early atypical lipidogram in overweight rats after 70 days of exposure is given in the Figure 6. What was the lipid profile at the end of the study? Can the authors add it? If not, why wasn't it also done at the end of the experiment?

Methods. In the methods, the number of rats (i.e. "n" value) used for individual determinations should be stated in this section.

Comments on the Quality of English Language

English is good, I have no comments about it.

Author Response

(The authors gave the same response as above.)

Round 2

Reviewer 2 Report

Comments and Suggestions for Authors

I thank the authors for modification of their manuscript. As mentioned before, it is difficult to understand what the new finding of this study really is. In the conclusion it seems that the main finding is that moderate overweight with preserved heart function is associated with pro-inflammatory phenotype. All other parts of the conclusion (given from line 639 on) come from other studies and the results are not in this manscuript (i.e. Na whole body regulation). So once again, what is the central hypothesis investigated in this study and what is the main result from this study? It is ok to discuss this outcome in the light of other studies from the authors but this cannot be the conclusion from this manuscript.   

Author Response

Please, see attached file. All responses are in a single document.

Reviewer 5 Report

Comments and Suggestions for Authors

Answer to Comment 1

In answer to Comment 1 you wrote that “The objective was to investigate whether inflammatory markers are present in the same subcellular compartments where ion-transporting ATPases and Ang II receptors are located. This objective is now stated in the RM (page 5, lines 162 to 165)”.

Lines 162-165 in revised manuscript: “Due to these observations, IL-6 and TNF-alpha can be considered ideal for studying inflammation during metabolic changes in HFD rats, and we have done this in the same preparation where ion-transporting ATPases and Ang II receptors are located.”

This sentence does not support the objective mentioned in the answer – investigation of presence (colocalization) of studied proteins in microsomal protein fraction.

The presented data related to inflammatory markers are not sufficient and do not support the stimulation of inflammation and conclusions related to inflammatory response written in lines 31-32 and lines 358-360.

Lines 31-32: “We conclude that being overweight causes inflammatory and ion transport cardiac alterations that could evolve into future heart dysfunction”.

Lines 358-360: “The existence of an intense inflammatory state that reaches the heart in HFD rats is confirmed by observations related to the increased levels of pro-inflammatory cytokines in the left-ventricular tissue (Figure 6), which were reversed by Ang-(3–4).”

Answer to Comment 5

In answer to this comment you wrote that hypertension was suppressed by Ang-(3–4) administration, as previously shown in Figure 1G from Crisóstomo et al. (2022) (ref. 30 in RM).

At the beginning of Discussion is summary of the main and novel findings of the present work. There is also sentence: “Hypertension in HFD rats is prevent by Ang-(3–4) administered at day 104 [30]..” (Lines 341-342).”

Figure 2 in revised manuscript does not show the effects of Ang-(3–4) administration on blood pressure. At the end of sentence is reference 30 – your paper published in 2022. Is the conclusion about the effects of Ang-(3–4) on blood pressure modulation in HFD rats based on these previously presented data (Fig. 1G) or on novel finding of the present work?

Author Response

(The authors gave the same response as above.)

Reviewer 6 Report

Comments and Suggestions for Authors

The authors modified the majority of the text of RM according to the my comments, but I still think that commenting on the results should not be in the Results section. For example, lines 82-83, 107-108, 110-111, 151 and 153-164. These parts should be in Discussion.

I requested that the authors document that GADPH was not affected by Ang-(3-4), (Figures 9A,B). The authors added lines (564-590) with the statement that "No differences were found within each group of membranes." However, the results of the 2-way ANOVA analysis are not shown. The results of the 2-way ANOVA analysis fot GADPH should be added as Supplementary Data.

Comments on the Quality of English Language

English is fine, minor corrections can be made during the publishing process (if the manuscript is accepted for publication).

Author Response

(The authors gave the same response as above.)

Round 3

Reviewer 2 Report

Comments and Suggestions for Authors

Dear Authors,

I thank for the changes in the manuscript that makes some points much clearer than before. However, I am still a little bit confused. You claim in the abstract (aims and conclusion) that ‘The central aim of this study was to investigate whether male Wistar rats chronically fed a 20 high-fat diet (HFD) during 106 days present high levels of interleukin-6 (IL-6) and tumor necrosis 21 factor-alpha (TNF-a) and Na+ and Ca2+ transport alterations in the left ventricle, together with 22 dyslipidemia and decreased glucose tolerance.’ and ‘We conclude that being overweight causes an increase in IL-6 and TNF-a and ion transport 33 alterations in the left ventricle that could evolve into future heart dysfunction.’ Therefore, neither in the aims nor in the conclusion Ang(3-4) does play any role. This is in sharp contrast to the title. So again, is the finding that Ang3-4 affects your parameters new or not and is this the aim of the study? Or is it a general description of the model. But then I do not understand why you share data with Ang3-4. I guess this is still unclear.

Author Response

We uploaded a condensed PDF file including: Letter to Editor, Response to Reviewer 2, marked version of the RM3 manuscript, and the unmarked version of the RM3 manuscript.

Reviewer 5 Report

Comments and Suggestions for Authors

The authors have included several required changes in the revised manuscript.

Author Response

Citation: "The authors have included several required changes in the revised manuscript."

Answer: We thank Reviewer 5 for his(her) comment.

Round 4

Reviewer 2 Report

Comments and Suggestions for Authors

I thank the authors for their reply.

Author Response

We thank the Reviewer.